# Adverse Events Associated with Universal versus Targeted Antifungal Prophylaxis among Lung Transplant Recipients—A Nationwide Cohort Study 2010–2019

**DOI:** 10.3390/microorganisms10122478

**Published:** 2022-12-15

**Authors:** Cornelia Geisler Crone, Signe Marie Wulff, Jannik Helweg-Larsen, Pia Bredahl, Maiken Cavling Arendrup, Michael Perch, Marie Helleberg

**Affiliations:** 1Centre of Excellence for Health, Immunity and Infections (CHIP), Copenhagen University Hospital Rigshospitalet, Blegdamsvej 9, 2100 Copenhagen, Denmark; 2Department of Infectious Diseases, Copenhagen University Hospital Rigshospitalet, Blegdamsvej 9, 2100 Copenhagen, Denmark; 3Department of Thoracic Anesthesia, Copenhagen University Hospital Rigshospitalet, 2100 Copenhagen, Denmark; 4Unit of Mycology, Statens Serum Institut, Artillerivej 5, 2300 Copenhagen, Denmark; 5Department of Clinical Microbiology, Copenhagen University Hospital Rigshospitalet, Blegdamsvej 9, 2100 Copenhagen, Denmark; 6Department of Clinical Medicine, University of Copenhagen, University Hospital of Copenhagen, Norre Allé 20, 2200 Copenhagen, Denmark; 7Department of Cardiology, Section for Lung Transplantation, Copenhagen University Hospital Rigshospitalet, Blegdamsvej 9, 2100 Copenhagen, Denmark

**Keywords:** transplantation, lung transplantation, fungal infections, aspergillus, prophylaxis, prevention, triazoles, adverse events, drug interactions, acute rejection

## Abstract

Background: Invasive fungal infections in lung transplant (LTX) recipients cause substantial morbidity, but the best strategy for prevention has not yet been determined. We evaluated adherence to and rates of adverse events of universal versus targeted prophylaxis. Methods: All LTX recipients in the Danish National LTX Centre (2010–2019) were included. Before July 2016, universal voriconazole prophylaxis was used. After July 2016, only high-risk patients received targeted prophylaxis with posaconazole and inhaled amphotericin B. Proportions of triazole discontinuation, side-effects, off-target calcineurin-inhibitor (CNI) levels, and acute rejection were compared between the two periods. Results: Universal and targeted prophylaxis was initiated in 183/193 and 6/102 patients, respectively. Only 37% completed > 9 of the intended 12 weeks of voriconazole; 72% of discontinuations were due to hepatotoxicity. In the universal vs. targeted prophylaxis period, 89% vs. 72% (*p* < 0.001) patients had low CNI episodes, and 37% vs. 1% (*p* < 0.001) of these were associated with discontinuation of triazole; 40% vs. 14% (*p* < 0.001) had acute rejection; and 23% vs. 3% (*p* < 0.001) had acute rejection associated with low CNI episodes. Conclusions: Universal voriconazole prophylaxis was associated with high rates of discontinuation, mainly caused by hepatotoxicity. In comparison to the targeted posaconazole period, more patients had low CNI levels and acute rejection in the universal voriconazole period.

## 1. Introduction

Invasive fungal infections (IFI) in lung transplant recipients (LTXr) are associated with high mortality [1,2]. The main strategies for the prevention of IFI after transplantation are universal or targeted prophylaxis, or pre-emptive therapy with systemic mould-active triazoles [3,4]. The benefit of antifungal prophylaxis is debated, and recent systematic reviews with meta-analyses did not find a convincing protective effect of antifungal prophylaxis on the prevention of IFI and reduction in associated mortality [5,6,7,8]. The use of triazoles is complicated by side-effects and drug–drug interactions [9,10]. In many lung transplant centers, voriconazole has been the drug of choice for antifungal prophylaxis [3,11]. Side-effects, primarily gastrointestinal upset, hepatotoxicity, and neurotoxicity, are common [9,12]. The newer triazole, posaconazole, formulated as a delayed release tablet (POS-Tab), was approved by European Medical Agency (EMA) in 2014 [12]. A recent study comparing itraconazole, voriconazole, and posaconazole for antifungal prophylaxis in LTXr demonstrated that posaconazole had fewer side-effects, and was less frequently discontinued, compared to voriconazole [13]. Triazoles interact with the metabolism of several drugs, including the immunosuppressive agents calcineurin-inhibitors (CNI), used for the prevention of graft rejection following transplantation. Triazole treatment increases CNI plasma levels. Dose reduction is recommended with CNI during triazole treatment to prevent elevated plasma levels of CNI and, potentially, over-immunosuppression and nephrotoxicity.

In the Danish National Lung Transplantation Centre, universal antifungal prophylaxis with voriconazole was the standard of care in the period of 2004–2016. The strategy for antifungal prophylaxis was changed in 2016 to targeted prophylaxis for high-risk patients with POS-Tab + inhaled liposomal amphotericin B. This change was motivated by a previous Danish study by Tofte et al., evaluating rates of *Aspergillus* infections before and after implementation of universal voriconazole prophylaxis. The study did not find a protective effect of this regimen when compared to a lack of prophylaxis [14]. Adherence and side-effects of voriconazole were not evaluated in the study. It is possible that the missing effect on the reduction in rates of *Aspergillus* infections after implementation of universal voriconazole prophylaxis was due to toxicity and poor adherence to the scheduled antifungal prophylaxis. A personalized approach for antifungal prophylaxis, with a better-tolerated antifungal regimen prescribed only for high-risk patients, may result in better outcomes. 

We studied proportions of prescription and completion of antifungal prophylaxis, causes of premature discontinuation, and triazole-associated adverse events, before and after the change in the Danish antifungal prophylaxis strategy.

## 2. Materials and Methods

### 2.1. Study Population

We included all Danish adult (>16 years) patients receiving lung transplantation 1.1.2010–31.12.2019 at the Danish National Lung Transplantation Centre, Copenhagen University Hospital Rigshospitalet. 

### 2.2. Data Sources

Data regarding transplantation and patient characteristics were retrieved from the national lung transplantation database. Results of pathological, microbiological, and biochemical examinations, performed as part of the clinical practice, were collected from nationwide registries through the Centre of Excellence for Personalized Medicine of Infectious Complications in Immune Deficiency (PERSIMUNE) Data Warehouse [15]. Data on prescription of and adherence to antifungal medication and side-effects were collected through review of medical records and organized in a RedCap database [16].

### 2.3. Definitions

Patients receiving < 75% (<9/12 weeks) of the intended duration were considered to have discontinued prophylaxis prematurely. This definition was chosen based upon clinical assessment, since no well-established definition exits in the literature. Side-effects leading to premature discontinuation of prophylaxis were reported by clinicians and recorded.

Adverse events were assessed by evaluating the selected laboratory tests, listed below, that were performed within the first 120 days after transplantation. Alanine aminotransferase (ALT), bilirubin, alkaline phosphatase, and creatinine measurements were graded according to Common Terminology Criteria for Adverse Events (CTCAE) criteria (grades 0–4 according to degree of elevation) [17], see Appendix A. Elevated biomarkers corresponding to CTCAE grade ≥ 2 were classified as high. 

Transbronchial biopsies were evaluated for rejection by specialized transplant pathologists, and were recorded as acute rejection requiring treatment when graded ≥ A2, according to ISHLT criteria [18]. 

An episode of high or low CNI plasma levels was defined as ≥2 consecutive plasma levels 33% above or below the CNI target range limits, respectively (target ranges available in Appendix C). An episode of low CNI was considered to be associated with discontinuation of triazole if it occurred 0–14 days from the day of discontinuation. Acute rejections were considered to be associated with a low CNI episode if they occurred 0–30 days from start of a low CNI episode. High creatinine was considered associated with a high CNI episode if occurring between two days prior to and seven days after the start of a high CNI episode. The off-target CNI episodes started at the first off-target measurement and ended at the subsequent measurement within the normal range. Patients who had ≥ 33% of their total CNI measurements in the low range (only including measurements taken prior to first acute rejection) were considered to have a high proportion of low CNI measurements, and are referred to as “ManyLowCNI” in the following sections.

### 2.4. Standard Protocols 

#### 2.4.1. Antifungal Prophylaxis

From July 2004 to July 2016, the use of universal prophylaxis with a voriconazole tablet 200 mg twice a day was recommended in the first 12 weeks after transplantation. In July 2016, the guideline was changed, recommending targeted prophylaxis for high-risk patients only (risk criteria available in Appendix B) with POS-Tab 300 mg once a day and inhalation liposomal amphotericin B 25 mg once a day from the time of transplantation to 12 weeks after transplantation. Therapeutic drug monitoring of the triazoles, when given as prophylaxis, was not routinely performed in either period.

#### 2.4.2. Immunosuppression and Other Prophylaxis

All patients received induction therapy with methylprednisolone and thymoglobuline, followed by maintenance therapy with a CNI, prednisolone, and an antiproliferative agent. The standard immunosuppressive protocol was changed in April 2017 due to participation in a randomized control multicenter study (ScanCLAD), with the aim of evaluating the effect of cyclosporine vs. tacrolimus on chronic lung allograft dysfunction. Initiated by this study, the preferred antimetabolite was changed from azathioprine to mycophenolat mofetil for all patients. Prior to the study, cyclosporine was the preferred calcineurin-inhibitor. During ScanCLAD enrollment, 57 patients were randomized to receive cyclosporine or tacrolimus. Details on the immunosuppression protocol and other antimicrobial prophylaxis guidelines are described in Appendix C.

#### 2.4.3. Routine Sampling

All patients were followed routinely with bronchoscopy, using bronchoalveolar lavage (BAL) sampling and transbronchial biopsies, at week one, two, six, and twelve, and at six, twelve, eighteen, and twenty-four months after transplantation during the study period. As is routine, BAL fluid was sent for microbiological examination by microscopy and culture. All BAL fluids and biopsies were sent for pathological examination, including Grocott–Gomori’s Methenamine Silver staining and microscopy. Additional bronchoscopy with BAL/biopsies or other respiratory tract sampling was performed upon clinical indication. The measurement of CNI plasma levels was performed upon clinical indication on a patient level, and this practice was not changed during the study period.

### 2.5. Statistics

Patient characteristics and side-effects were assessed by descriptive statistical analyses. To evaluate the significance of the differences of categorical variables, Chi^2^ or Fisher’s exact test were used, when appropriate. Continuous variables were compared using the Wilcoxon signed-rank test. Associations between prophylaxis periods, ManyLowCNI, and acute rejections were evaluated using uni-, bi-, and multi-variable logistic regression analyses, adjusted for sex, age, and type of CNI. Due to potential collinearity between prophylaxis regime periods and low levels of CNI, a combined variable was created, categorizing patients by prophylaxis regime period and including ManyLowCNI or not in the groups: “Targeted prophylaxis and not ManyLowCNI”; “Targeted prophylaxis and ManyLowCNI”; “Universal prophylaxis and not ManyLowCNI”; “Universal prophylaxis and ManyLowCNI”. None of the 14 patients in the “Targeted prophylaxis and ManyLowCNI” group had rejection, so the two targeted prophylaxis groups were merged into one group: “Targeted prophylaxis”. These analyses on rejection as an outcome were performed post hoc, due to the unexpected finding of more episodes of acute rejection in the universal prophylaxis period. 

All analyses were computed at a two-sided α level of 5% with R software, version 3.6.1.

The study was approved by the Danish National Board of Health (3–3013–1060/1/ approved 19 March 2020) and the Danish Data Protection Agency (RH-2016–47; approved 16 January 2019).

## 3. Results

### 3.1. Patient Characteristics

We included 295 LTXrs, of which 193 received LTX during the universal antifungal prophylaxis period (2010–2016) and 102 during the targeted antifungal prophylaxis period (2016–2019).

The median age, in the total cohort, was 53 years (IQR 43–58), and the median body mass index (BMI) was 21.8 (IQR 18.6–26.0). Age and BMI at time of transplantation were higher among patients transplanted during the period of targeted prophylaxis, and a higher proportion of patients received double lung transplant compared to the universal prophylaxis period (Table 1). The main underlying disease leading to transplantation was emphysema in both periods, but a larger percentage of LTXr had cystic fibrosis in the universal prophylaxis period compared to the targeted period (19% vs. 8%, *p* = 0.013). Patient characteristics are summarized in Table 1.

### 3.2. Premature Discontinuation

In the universal prophylaxis period, 183 of the 193 (95%) LTXrs initiated voriconazole prophylaxis per the protocol (Table 2). In 2016–2019, 6 of 102 (6%) received targeted prophylaxis with POS-Tab and amphotericin B inhalations, as not all patients qualified for targeted prophylaxis per protocol by fulfilling the IA high-risk criteria. 

Among the 183 patients receiving voriconazole prophylaxis, 114 (62%) discontinued prophylaxis prematurely. The median time receiving voriconazole prophylaxis was 36 days (IQR 12–84). Among those who discontinued voriconazole prematurely, the median time to discontinuation was 15 days (IQR 7–62). The main cause of premature discontinuation was hepatotoxicity, which was reported in 82 (72%) of the LTXr who discontinued voriconazole (Figure 1). 

Universal voriconazole prophylaxis was paused in 36 patients. After resumption of voriconazole, 23/36 (64%) patients completed the voriconazole prophylaxis course as per the protocol. 

### 3.3. Adverse Events

#### 3.3.1. Hepatotoxicity

During the first 120 days after transplantation, a higher proportion of LTXr in the universal vs. targeted prophylaxis period had ≥1 episode of high ALT (30% vs. 11%, *p* < 0.001) and alkaline phosphatase (42% vs. 12%, *p* < 0.001, Table 2).

In the group receiving universal voriconazole prophylaxis, plasma levels of ALT peaked after the recent discontinuation of voriconazole, and dropped to baseline levels at >14 days after discontinuation (Figure 2). 

#### 3.3.2. High CNI Plasma Levels

During the first 120 days after transplantation, the total number of CNI measurements taken per patient was similar in the two prophylaxis periods, with a median of 38 samples per patients in both periods (number of measurements per patient displayed in Appendix D). The proportions of LTXr with ≥1 episode of elevated CNI plasma level were comparable in the universal and the targeted prophylaxis periods (22% vs. 28%, *p* = 0.26). The proportions of LTXr with creatinine elevation were also comparable in the two periods (Table 2). No significant difference was found in the proportion of LTXrs with elevated creatinine associated in time with a high CNI episode (4% vs. 9%, *p* = 0.17) when comparing the universal vs. the targeted period, respectively.

#### 3.3.3. Low CNI Plasma Levels

More patients had ≥1 episode of low CNI plasma levels in the universal prophylaxis period vs. the targeted prophylaxis period (89% vs. 72%, *p* < 0.001). The median number of low CNI episodes per patient was two (IQR 1–3, range 0–12) and one (IQR 0–2, range 0–10) among patients in the universal and targeted prophylaxis period, respectively. The median accumulated amount of time for which patients had low CNI levels was 8.1 days (IQR 2.2–20.2) for the total study period, and 11 days (IQR 4–24) and 3 days (IQR 0–9), *p* < 0.001, in the universal and targeted prophylaxis period, respectively. During the universal prophylaxis period, 71 (37%) patients had a low CNI episode occurring 0–14 days after voriconazole discontinuation (Table 2).

#### 3.3.4. Acute Rejections

The proportion of LTXr with ≥1 acute rejection episode was higher in the universal prophylaxis period compared to the targeted (40% vs. 14%, *p* < 0.001). In the universal prophylaxis period, the median number of acute rejections per patient was zero (IQR 0–1, range 0–4) and this number was also zero (IQR 0–0, range 0–2) in the targeted prophylaxis period. The proportion of acute rejections in relation to a low CNI episode was higher in the universal vs. the targeted period (23% vs. 3%, *p* < 0.001).

In the post hoc logistic regression models, the univariable odds ratio (OR) of acute rejection was 4.26 (95% CI 2.32–8.31) for the universal compared to the targeted prophylaxis period, and 1.80 (95% CI 1.03–3.12) when comparing patients with ManyLowCNI to those without ManyLowCNI. In multivariable analyses, the odds of acute rejection were higher among patients both with and without ManyLowCNI in the universal prophylaxis period compared to the targeted period (Table 3). When ManyLowCNI was taken into account for the patients in the universal period, the risk was higher in the subgroup with ManyLowCNI than in the subgroup without ManyLowCNI, OR 5.22 (95% CI 2.37–11.9), and OR 3.14 (95% CI 1.58–6.61), respectively (Table 3).

## 4. Discussion

In this study, which included 295 LTXr during two time periods with different antifungal prophylaxis protocols, we found that a large proportion of patients discontinued voriconazole prophylaxis prematurely. The main reason for premature discontinuation of voriconazole was hepatoxicity. More patients had episodes of low plasma levels of CNI and acute rejections during the period with universal vs. targeted antifungal prophylaxis, and a large proportion of low CNI episodes was related, in time, to voriconazole discontinuation.

The high proportion of premature discontinuations of voriconazole in our study (62%) confirms the findings of previous, smaller single-center studies, reporting premature discontinuation proportions of 41% [19], 69% [13], and 84% [20]. However, some studies reported lower discontinuation proportions of 14% (N = 65) [21], 34% (N = 35) [22], and 27% (N = 93) [23]. Several factors may have contributed to these differences, such as variations in tolerability, underlying diseases, and concomitant medications, as well as differences in voriconazole metabolization related to genetic disposition [24].

In the present study, the predominant side-effect of voriconazole was hepatotoxicity, which is a known and consistent problem in LTXr. However, the proportion of discontinuations due to hepatotoxicity in this study was high when compared to previous studies [20,21,22,23]. Risk factors for voriconazole-related hepatotoxicity in LTXr have been identified by Luong et al., who found cystic fibrosis and use of azathioprine to be associated with hepatoxicity [25]. A relatively high proportion of our study population had cystic fibrosis, especially in the universal prophylaxis period, in which azathioprine was also used for all patients per protocol. 

Therapeutic drug monitoring (TDM) of voriconazole, when given as prophylaxis, was not a part of the standard protocol in our center throughout the study period. Mitsani et al. investigated voriconazole trough levels in a LTXr cohort (N = 93) with and without side-effects, and did not find a correlation between elevated voriconazole levels and nausea, CNS toxicity, or liver enzyme elevation [23]. However, a meta-analysis found that patients with supratherapeutic voriconazole levels had an almost four-fold risk of hepatotoxicity when data on hematological and solid organ transplant cohorts were pooled from 11 studies [26]. A recent study on TDM of voriconazole treatment in lung transplant recipients also showed that 82% of patients with a plasma level above 2.13 μg/mL had hepatotoxicity [27]. Toxic levels of voriconazole could have contributed to the high proportion of side-effects and patients discontinuing voriconazole in our study, despite the relatively low dosage.

Differences in protocols of co-administered medication, causing similar side-effects or possibly reinforcing the voriconazole side-effects, could also affect adherence to voriconazole. In our center, voriconazole was frequently discontinued in parallel/”en bloc” with other medications when patients experienced side-effects, which complicates the ascertainment of which agents caused side-effects. The clinical readiness to resume antifungal prophylaxis after pausing due to side-effects can also differ between centers. We found that resumption of voriconazole prophylaxis after pause was successful in the majority of patients where voriconazole was reassumed (23/36 patients).

The high proportion of premature discontinuations has likely influenced the results in the previous study from our center by Tofte et al., who evaluated the protective effect of voriconazole prophylaxis on IFI when comparing universal voriconazole prophylaxis to no prophylaxis [14]. The study did not find a preventive effect of voriconazole, but all patients initiating voriconazole were considered to have completed the prophylaxis per protocol [14]. The absence of TDM during voriconazole prophylaxis in our institution may also have contributed to this lack of preventive effect on IFIs, which was demonstrated by Tofte et al.

We found that more patients in the universal compared to the targeted prophylaxis group had episodes of low CNI levels and acute rejections. 

A high proportion of episodes with low CNI plasma levels was associated in time to voriconazole discontinuation, which could indicate lack of increase in CNI dosage upon discontinuing voriconazole. This might be related to the clinical setting/situation when voriconazole is discontinued due to side-effects, e.g., patients discontinuing on their own initiative without consulting a physician, or through telephone consultation. In these situations, it can be more challenging to secure timely CNI dosage adjustments and parallel CNI therapeutic drug monitoring. 

We also found that many acute rejections were associated with low CNI episodes, more frequently seen during the universal voriconazole prophylaxis period. However, many factors could potentially have affected the difference in acute rejections observed between the two prophylaxis periods. A change in protocol for immunosuppressive regimes was made in 2017, due to participation in a Scandinavian randomized controlled study. This included a change in center protocols from azathioprine to mycophenolat mofetil to all patients. Further, 57 LTXrs were randomized to cyclosporine or tacrolimus. Reports of these immunosuppressants’ effect on the prevention of acute rejection are ambiguous [28], but these changes could also affect the differences in acute rejections over the two periods. Although not the primary aim of this study, we investigated the associations between prophylaxis periods and acute rejections in post hoc multivariable analyses. We found that the association between the universal prophylaxis regime and increased rejection remained strong after adjustment for potential confounders, including immunosuppressive drugs. The analysis in which patients in the universal prophylaxis period were stratified into groups with and without ManyLowCNI indicated that the low levels of CNI could, in part, explain the increased rejections, but also that the universal period still seemed independently associated with increased rejection. This could be related to a deficient classification of low CNI levels, to other voriconazole-related mechanisms leading to rejection, or to unmeasured confounding.

Our study has some other important limitations. Not all patients who qualified for prophylaxis by fulfilling the IA high-risk criteria, which are defined by the guidelines, were started on targeted prophylaxis with POS-Tab and inhaled amphotericin B. Due to the small number of patients starting targeted prophylaxis, we were unable to evaluate discontinuation proportions and side-effects of this regime, as initially intended. Differences in characteristics of the study population in the two periods may contribute to some of the observed differences in adverse outcomes. Strengths of the study include the rather large study population and the use of nationwide data registries linked using their Danish civil registration numbers, allowing almost complete data availability for healthcare contacts, pathological and laboratory results, and for patients who attended follow-up appointments at other hospitals. 

Our study adds quantitative results regarding adverse events that are important to the debate on costs and benefits of antifungal prophylaxis. The findings raise awareness of potential adverse events due to drug–drug interactions during universal antifungal voriconazole prophylaxis, which may have important clinical implications. Previous studies from our center did not find lower rates of IFI with universal prophylaxis. The high discontinuation rates demonstrated that including adherence to prophylaxis on a patient level, when studying effectiveness of antifungal prophylaxis, may improve our understanding of the challenges of antifungal prophylaxis in lung transplant recipients. An updated evaluation comparing rates of IFI during different prophylactic strategies and adherence is underway.

## 5. Conclusions

In summary, we found that the proportion of premature discontinuations of voriconazole prophylaxis was high, mainly due to hepatic side-effects. Patients with episodes of low plasma levels of CNI and acute rejections were more frequent during the period with universal versus targeted antifungal prophylaxis. A large proportion of patients had low CNI episodes that were associated in time with voriconazole discontinuation, as well as acute rejection episodes that were associated in time with low CNI episodes. This underlines the challenges and the low adherence regarding the use of voriconazole prophylaxis in the lung transplant population, in addition to the importance of frequent monitoring and dose adjustment of CNI when co-administered with voriconazole.

## Figures and Tables

**Figure 1 microorganisms-10-02478-f001:**
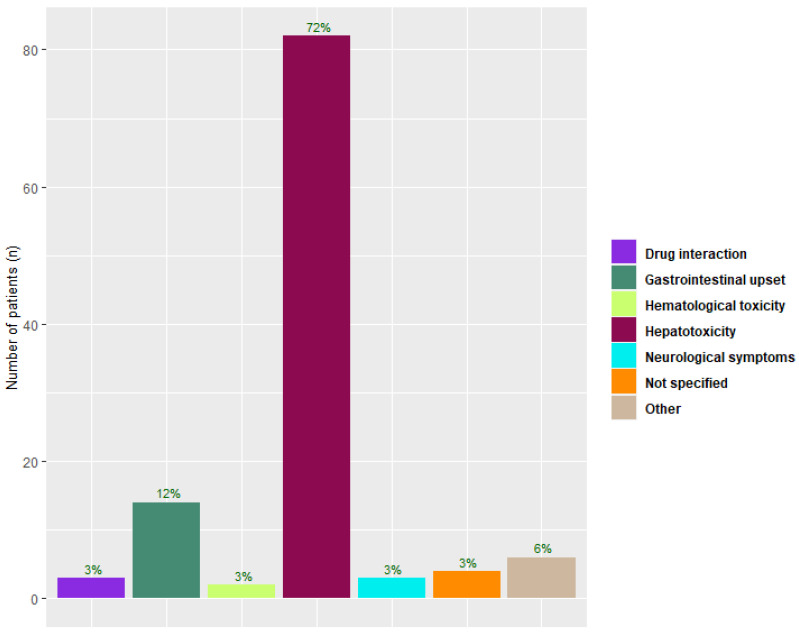
Cause of premature discontinuation of universal voriconazole prophylaxis among lung transplant recipients. Bar plot displays the distribution of side-effects leading to discontinuation as reported by clinicians. Percentage of side-effects shown are proportions of all patients discontinuing universal voriconazole prophylaxis.

**Figure 2 microorganisms-10-02478-f002:**
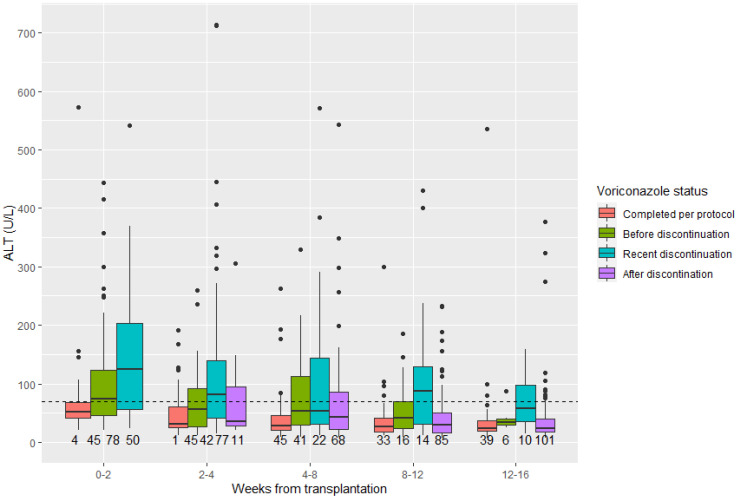
Alanine aminotransferase levels in lung transplant recipients initiating universal voriconazole prophylaxis. Box plots with peak values of alanine aminotransferase (ALT) per patient in each time period are grouped according to voriconazole status. Peak ALT values for each patient were recorded in time intervals after transplantation and were categorized according to the status of voriconazole prophylaxis. Values in patients who had started voriconazole and completed it per protocol were grouped as “Completed per protocol”. The values of patients who, at some point, discontinued prophylaxis prematurely, but were still on voriconazole in the current time period, were categorized as “Before discontinuation”. Those who had discontinued voriconazole within a 14-day period were categorized as “Recent discontinuation”, and those who had discontinued voriconazole more than 14 days earlier as “After discontinuation”. The number of observations per group is shown below each box plot. Dashed line indicates male upper limit of normal (70 U/L).

**Table 1 microorganisms-10-02478-t001:** Patient characteristics at time of lung transplantation in the universal- and targeted antifungal prophylaxis periods.

	Universal Prophylaxis Period (n = 193)	Targeted Prophylaxis Period (n = 102)	*p*-Value
Male, *n (%)*	106 (55)	46 (45)	0.14
Age, *median (IQR)*	52 (42, 57)	55 (45, 58)	0.04
BMI, median *(IQR)*	21 (18, 25)	23.3 (20, 28)	0.01
**Underlying Disease, n (%)**			
Cystic fibrosis	36 (19)	8 (8)	0.01
Emphysema	84 (44)	52 (51)	0.27
Primary pulmonary hypertension	5 (3)	5 (5)	0.32
Pulmonary fibrosis	50 (26)	33 (32)	0.30
Retransplantation	6 (3)	1 (1)	0.43
Sarcoidosis	12 (6)	3 (3)	0.22
**Type of Lung Transplant, n (%)**			
Double	168 (87)	97 (95)	
Single	25 (13)	5 (5)	0.048

Universal antifungal prophylaxis period: voriconazole was given three months following transplantation for all patients. Targeted antifungal prophylaxis: systemic posaconazole and inhaled amphotericin B were administered three months following transplantation for high-risk patients. N = number of patients, IQR = interquartile range, BMI = body mass index.

**Table 2 microorganisms-10-02478-t002:** Number (%) of lung transplant recipients with adverse events during the first 120 days after transplantation in the universal- and targeted antifungal prophylaxis periods.

	Universal Prophylaxis Period	Targeted Prophylaxis Period	*p*-Value
Initiated prophylaxis	183 (95)	6 (6)	<0.001
Completed prophylaxis	69/183 (38)	4/6 (67)	0.22
High ALT	56 (30)	11 (11)	<0.001
High alkaline phosphatase	80 (42)	12 (12)	<0.001
≥1 episode of low CNI	172 (89)	73 (72)	<0.001
ManyLowCNI	59 (31)	14 (14)	0.002
Low CNI episode related to triazole stop	71 (37)	1 (1)	<0.001
Acute rejection	78 (40)	14 (14)	<0.001
Acute rejection related to low CNI episode	44 (23)	3 (3)	<0.001
≥1 high of CNI episode	42 (22)	29 (28)	0.26
High creatinine	87 (45)	54 (53)	0.24
High creatinine related to high CNI episode	8 (4)	9 (9)	0.17

Universal antifungal prophylaxis period: Voriconazole given three months following transplantation for all patients. Targeted antifungal prophylaxis: systemic posaconazole and inhaled amphotericin B three months following transplantation for high-risk patients. High biomarker indicating patients with ≥1 episode of grade ≥ 2 elevation according to Common Terminology Criteria for Adverse Events criteria. ManyLowCNI ≥ 33% of total CNI measurements at low level, ALT = alanine aminotransferase, CNI = calcineurin-inhibitor, acute rejection = pathological grading ≥ A2.

**Table 3 microorganisms-10-02478-t003:** Post hoc analyses on factors associated with acute rejection.

	Rejection N = 92, n (%)	No Rejection N = 203, n (%)	Univariable OR (95% CI)	Model 1 OR (95% CI)	Model 2 OR (95% CI)
Tacrolimus	4 (4)	29 (14)	Ref.	Ref.	Ref.
Cyclosporine	88 (96)	174 (86)	3.67 (1.39–12.7)	1.82 (0.61–6.76)	1.89 (0.62–7.08)
**Prophylaxis regime** **and ManyLowCNI status**					
Targeted prophylaxis period	14 (15)	88 (43)	Ref.	Ref.	Ref.
Universal prophylaxis period without ManyLowCNI	48 (52)	86 (42)	3.51 (1.84–7.03)	3.05 (1.54–6.38)	3.14 (1.58–6.61)
Universal prophylaxis period with ManyLowCNI	30 (33)	29 (14)	6.50 (3.09–14.3)	5.79 (2.69–13.0)	5.22 (2.37–11.9)

Univariable: univariable logistic regression. Model 1: bivariable logistic regression model including combined variable “Prophylaxis regime + ManyLowCNI” and calcineurin-inhibitor. Model 2: multivariable logistic regression models adjusted for sex, age, and calcineurin-inhibitor. ManyLowCNI = patients with > 33% of total CNI measurements > 33% below target. CNI = calcineurin-inhibitor, OR = odds ratio, CI = confidence interval. Targeted prophylaxis: all patients in the targeted prophylaxis period were pooled in this group regardless of ManyLowCNI status, since 0/14 patients, initially grouped as “Targeted prophylaxis and ManyLowCNI”, had rejection.

## Data Availability

The data sets contain sensitive patient data governed by General Data Protection Regulation and Danish law. Because of Danish legislation and approvals granted by the Danish Data Protection Agency, it is not possible to upload raw data to a publicly available database. However, access to these data can be made available from the corresponding author upon reasonable request, provided a relevant data processing agreement is entered into according to current regulations.

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
