# Peer review of "Adverse Events Associated with Universal versus Targeted Antifungal Prophylaxis among Lung Transplant Recipients—A Nationwide Cohort Study 2010–2019"

_microorganisms, 2022, doi:10.3390/microorganisms10122478_

Round 1

Reviewer 1 Report

Dear Authors:

It is necessary to know which pathogens (Fungi) were present in each of the invasive fungal infections, as this gives a better picture of the effectiveness of the azole being used.  The authors should know that not all fungi are susceptible to voriconazole and neither to posoconazole or the combination of these with liposomal amphotericin B, this is very clear between mucorales and aspergillosis, so this limits the study because if we knew the causative agent and the azole that was prescribed, we could exclude cases that do not meet the study criteria.

If the authors can add to this study the pathogens and azole used in the study, this would help to improve the work and better understand the results.  

Reviewer 2 Report

the manuscript covers an interesting topic aimed to assess edverse events associated with universal versus targeted anti-fungal prophylaxis among lung transplant recipients in a danish cohort.

The title is well representative of the content of the manuscript.

in the abstract, bullet points are not needed. please revise the abstract according to the journal instructions.

introduction provided a sufficient level of background information.

methods are well-described 

results are clearly reported, and tables and figures are self-explicative.

The discussion offers a good interpretation of the results, however, in the introduction a brief presentation of the main findings should be added before comparing results with previous evidence.

In the conclusions, please add a public health considerations/implications of these results

Round 2

Reviewer 1 Report

I considerar that the authors have clarifica my questions correctly